# The Prognostic and Predictive Value of Body Mass Index in Patients with HR+/HER2− Breast Cancer Treated with CDK4/6 Inhibitors: A Systematic Literature Review

**DOI:** 10.3390/cancers18010081

**Published:** 2025-12-26

**Authors:** Larisa Maria Badau, Cristina Marinela Oprean, Andrei Dorin Ciocoiu, Paul Epure, Brigitha Vlaicu

**Affiliations:** 1Discipline of Hygiene, “Victor Babes” University of Medicine and Pharmacy, Eftimie Murgu Square No. 2, 300041 Timisoara, Romania; vlaicu@umft.ro; 2Department of Oncology, ONCOHELP Hospital Timisoara, Ciprian Porumbescu Street, No. 59, 300239 Timisoara, Romania; cristina.oprean@umft.ro; 3Department of Oncology, ONCOMED Outpatient Unit Timisoara, Ciprian Porumbescu Street, No. 59, 300239 Timisoara, Romania; 4ANAPATMOL Research Center, “Victor Babes” University of Medicine and Pharmacy, 300041 Timisoara, Romania; 5Department of Microscopic Morphology, “Victor Babes” University of Medicine and Pharmacy, 300041 Timisoara, Romania; andrei.ciocoiu@umft.ro; 6Department of Oncology, City Clinical Emergency Hospital of Timisoara, “Victor Babes”, Blvd. No. 22, 300595 Timisoara, Romania; 7“Pius Brinzeu” County Emergency Hospital, 300723 Timisoara, Romania; epurepaul1297@gmail.com

**Keywords:** breast cancer, CDK 4/6 inhibitors, BMI, progression-free survival

## Abstract

This systematic review evaluated the association between BMI and clinical outcomes in patients with HR+/HER2− MBC receiving CDK4/6i. The 14 eligible studies yielded heterogenous results; while several investigations identified a prognostic advantage for overweight or obese patients, others failed to demonstrate a significant predictive correlation between BMI and survival outcomes. Current evidence indicates that BMI is an insufficient surrogate for body composition and does not consistently predict CDK4/6i efficacy. Further research integrating detailed anthropometric and metabolic profiling is needed to clarify the role of body composition in HR+/HER2− MBC.

## 1. Introduction

Breast cancer (BC) remains the leading cancer diagnosis among women globally. Data from the Global Cancer Observatory (2022) report over 2.2 million new cases and roughly 660,000 deaths [1]. Notably, nearly 70% of the reported cases are classified as hormone receptor-positive (HR+)/human epidermal growth factor receptor 2-negative (HER2−) [2].

Recent advances in identifying prognostic and predictive factors have shifted therapeutic approaches. Body mass index (BMI) is becoming an interesting area of study aimed at enhancing the precision and effectiveness of BC management, particularly estrogen receptor-positive disease [3,4].

In contrast to early-stage BC, recent evidence in the metastatic setting suggests that obesity may paradoxically serve as a protective factor [5]. Nevertheless, the prognostic influence of a high BMI remains controversial, as retrospective data and institutional case series have yielded inconsistent results [6,7,8,9,10]. This lack of consensus is largely due to the methodological heterogeneity across studies, including variations in cohort size, treatment regimens, BMI thresholds and clinical endpoints.

The introduction of CDK4/6 inhibitors (CDK4/6i) (palbociclib, ribociclib and abemaciclib) has fundamentally shifted the management of HR+/HER2− MBC. The efficacy and safety profiles of these agents have been proven through the landmark PALOMA, MONALEESA and MONARCH trials, as well as through real-world experience [11,12,13,14,15,16]. The utility of these drugs lies in how they disrupt the G1 phase progression. Unlike other cyclin-dependent kinases (CDKs) such as CDK1 and CDK2, which regulate multiple phases of the cell cycle, CDK4 and CDK6 play a pivotal role in driving the G1-to-S phase transition by phosphorylation of the retinoblastoma (Rb) protein. In MBC, where this regulatory checkpoint is frequently disrupted, CDK4/6i can effectively reestablish cell cycle control [17,18]. This targeted approach has moved beyond clinical trials and has become the standard of care.

Preclinical data have redefined CDK4 and CDK6 as key metabolic switches that control adipogenesis, lipid synthesis, glucose regulation and muscle metabolism [17,18,19]. Because CDK4 is essential for fat cell maturation independent of cell division, its inhibition may directly reshape body composition [19]. Recent research has further proposed these kinases as targets for obesity intervention, implying that CDK4/6i therapy may directly impact both adipose and skeletal muscle mass [20,21].

Given that CDK4/6i actively modulate lipogenesis and interact with obesity-driven metabolic pathways, understanding the influence of BMI on clinical outcomes in HR+/HER2− MBC is warranted [22]. This systematic review evaluates the current evidence regarding the association between BMI and survival in HR+/HER2− MBC patients receiving combined CDK4/6i and endocrine therapy (ET). By synthesizing these findings, we aim to clarify whether this accessible metric can serve as a meaningful predictor of treatment success in a complex metabolic landscape. Additionally, studies conducted in early-stage disease were included to provide contextual insights and to explore potential differences between early and metastatic settings. 

## 2. Materials and Methods

This systematic review was conducted in accordance with the 2020 Preferred Reporting Items for Systematic Reviews and Meta-Analyses (PRISMA) guidelines [23]. The review was not prospectively registered in a public database. The PICO framework—Population, Intervention/Exposure, Comparison, and Outcome—was applied as outlined below:Population (P): Patients diagnosed with HR+/HER2− MBC;Intervention (I): Treatment with CDK4/6i (palbociclib, ribociclib, or abemaciclib) in combination with ET;Comparison (C): Comparisons across different BMI categories;Outcomes (O): Prognostic and predictive impact of BMI on treatment efficacy, assessed by progression-free survival (PFS), overall survival (OS), and response rate (RR).

A systematic literature search was performed using the PubMed and Scopus databases, covering studies published between 2015 and 2025. These sources were selected to ensure the inclusion of both high-quality and peer-reviewed biomedical literature. The study utilized Zotero software (version 7) as an automation tool to collect, compile, and organize all search results exported from the databases over the predefined search period.

The search strategy was developed to ensure high sensitivity in identifying studies evaluating BMI or related parameters in relation to outcomes of CDK4/6i therapy in HR+/HER2− MBC. A combination of Medical Subject Headings (MeSH) terms and free-text keywords was used to construct the search. Keywords included: (“breast cancer”) and (“palbociclib” or “ribociclib” or “abemaciclib” or “cdk4/6 inhibitor*” or “cyclin-dependent kinase 4 and 6 inhibitor*”) and (“body mass index” or “BMI” or “obesity” or “body weight” or “body composition”) and (“survival” or “progression-free survival” or “overall survival” or “treatment outcome” or “disease progression” or “response rate” or “objective response” or “clinical benefit” or “predictor*” or “predicting” or “responses” or “predictive” or “prognostic”).

Eligible studies were selected based on the following inclusion criteria:Studies including adult patients with HR+/HER2− MBC treated with CDK4/6i in combination with ET;Studies that evaluated BMI or other body composition markers, reported either as categorical groups or continuous variables;Studies that reported clinical outcomes such as PFS, OS, disease-free survival (DFS), and RR related to BMI or other markers;Full text articles available in English and published between 2015 and 2025;Studies involving human participants, specifically observational or randomized trials.

Exclusion criteria included:Studies not focused on HR+/HER2− MBC or not evaluating the association between BMI and outcomes related to CDK4/6i treatment;Publications reporting overlapping data;Articles published in languages other than English;Studies available only as abstracts, without accessible full-text;Publications presented in non-eligible formats, such as letters to the editor, case reports, editorials, conference proceedings, systematic reviews, or meta-analyses.

Although studies enrolling patients with HR+/HER2− MBC treated with CDK4/6i were prioritized, two included studies were conducted in early-stage disease to provide contextual comparison. However, conclusions regarding early-stage disease remain limited, as CDK4/6i were initially developed and extensively evaluated in the metastatic setting, where the majority of available evidence has been generated.

Titles and abstracts were screened independently by two reviewers. Discrepancies were resolved through mutual discussion. Full-text versions of potentially eligible studies were reviewed based on the predefined inclusion criteria. Data were extracted using a standardized table. Extracted information included study design, population characteristics, sample size, name of CDK4/6i, metrics, survival outcomes, hazard ratio (HR), *p*-value and statistical methods. Selected studies were assessed for quality and risk of bias, applying the Newcastle–Ottawa Scale (NOS) (Table 1). 

## 3. Results

A total of 57 records were identified through database searching (PubMed, Scopus), with 12 additional studies included from further citation snowballing. Following the removal of 12 duplicates and four records flagged by automated tools, 41 records were screened based on their titles and abstracts. Of these, 29 full-text articles were assessed for eligibility according to the inclusion and exclusion criteria. After curation, 14 studies [9,24,25,26,27,28,29,30,31,32,33,34,35,36] were included in the final systematic review, as illustrated in the PRISMA flow diagram (Figure 1).

The main characteristics of the 14 included studies evaluating the impact of BMI and some body composition variables are summarized in Table 2. Included studies were published between 2020 and 2024 across multiple countries and consisted of eight retrospective cohort studies, three prospective observational studies, one ambispective observational study and two subgroup analyses from clinical trials. The outcomes primarily focused on PFS and OS, while some studies also reported objective response rate (ORR), clinical benefit rate (CBR), and invasive disease-free survival (iDFS).

Of the included studies, four focused exclusively on palbociclib, one on ribociclib and two on abemaciclib. Additionally, three analyzed cohorts treated with either ribociclib or palbociclib, while two evaluated all three CDK4/6i. In the other two instances, the specific type of CDK4/6i was not specified.

BMI was analyzed either as a continuous variable or according to WHO categories (underweight < 18.5 kg/m^2^, normal 18.5–24.9 kg/m^2^, overweight 25–29.9 kg/m^2^, obese ≥ 30 kg/m^2^).

### 3.1. BMI and Improved Survival Outcomes

Four studies reported that higher BMI was associated with superior survival in the HR+/HER2− MBC population receiving CDK4/6i. Notably, Roncato et al. (2023) [24] found that patients with a BMI < 25 kg/m^2^ experienced significantly shorter PFS than those with a BMI ≥ 25 kg/m^2^ (HR = 2.32, 95% CI: 1.09–4.95; *p* = 0.0332). The authors also noted that lower BMI (<25 kg/m^2^) was a predictor for dose-limiting toxicities, leading to reduced treatment adherence and shortened survival. This finding suggests that overweight and obese patients may experience better survival outcomes compared with normal-weight individuals.

Çağlayan et al. (2024) [28] also stratified patients with HR+/HER2− MBC into three distinct categories based on BMI: normal weight (BMI < 25 kg/m^2^), overweight (25 kg/m^2^ ≤ BMI < 30 kg/m^2^), and obese (BMI ≥ 30 kg/m^2^). The study found that patients in the overweight category had the longest PFS and better RR, whereas both normal weight and obese patients demonstrated shorter outcomes, indicating a potential prognostic benefit linked to the overweight range. Specifically, the median PFS was 11.1 months (9.7–12.6) in obese patients compared with 9.3 months (5.3–13.4) in normal-weight patients, showing a statistically significant association between higher BMI and prolonged PFS (*p* = 0.02).

In the study by Chen et al. (2024) [31], conducted on an Asian population with HR+/HER2− MBC, a higher BMI was associated with a reduced risk of progression (HR = 0.943, 95% CI: 0.907–0.980; *p* = 0.003), supporting the hypothesis that a BMI ≥ 25 may confer a survival advantage in this setting.

In the real-world multicenter study by Shen et al. (2022) [35], conducted in 190 Chinese patients with HR+/HER2− MBC treated with palbociclib plus ET, higher BMI was associated with better PFS (HR = 0.98, 95% CI: 0.66–1.47, *p* = 0.928), emphasizing the hypothesis that increased body weight may be associated with improved treatment outcomes in this population.

### 3.2. No Significant Association Between BMI and Survival Outcomes

Although several studies highlighted a prognostic value for overweight or obese patients treated with CDK4/6i, other studies did not demonstrate a significant predictive or prognostic association between BMI and survival outcomes.

In contrast to studies conducted in the metastatic setting, Pfeiler G. et al. (2023) [25] performed a preplanned BMI-subgroup analysis of the PALLAS trial, evaluating the impact of BMI in patients with early-stage HR+/HER2− BC receiving adjuvant ET with or without palbociclib. The analysis found no impact of BMI on improvement in iDFS in the three categories of patients: normal weight (HR = 0.84, 95% CI: 0.63–1.12); overweight (HR = 1.10, 95% CI: 0.82–1.490); obese HR = 0.95, 95% CI: 0.69–1.30).

The study conducted by Franzoi et al. (2022) investigated the impact of BMI in patients with early-stage HR+/HER2− breast cancer treated with neoadjuvant endocrine therapy plus abemaciclib within the NEOMONARCH trial. Among 222 patients, no significant association was found between BMI and Ki-67 % change, clinical response (*p* = 0.261), or radiological response (*p* = 0.366) [27]. These results suggest that baseline BMI did not influence treatment efficacy in the neoadjuvant setting. Together, these two studies indicate that, in early-stage HR+/HER2− BC, BMI does not appear to significantly affect response or survival outcomes in patients receiving CDK4/6i in combination with ET.

In a pooled analysis of the MONARCH 2 and MONARCH 3 trials, Franzoi et al. (2021) [9] found that BMI did not significantly influence PFS (HR = 1.03; *p* = 0.81). However, BMI was significantly associated with treatment response; patients with an elevated BMI (≥25 kg/m) had a lower objective response rate (ORR) than their normal or underweight counterparts (OR = 0.73, 95% CI: 0.54–0.99; *p* = 0.04). The clinical CBR did not differ significantly between BMI categories (93.9% vs. 91.8%; OR = 0.73, 95% CI: 0.41–1.31; *p* = 0.28). These results indicate that BMI did not influence survival outcomes or overall clinical benefit, although it may be associated with variations in RR.

Lammers et al. (2023), in their real-world study using data from the SONABRE registry, did not find any differences between the weight categories regarding OS and PFS. Interestingly, the underweight patients (BMI < 18.5 kg/m^2^) tended to have poorer OS, although this difference did not reach statistical significance (HR = 1.45, 95% CI: 0.97–2.15; *p* = 0.07) [26].

In the study by Knudsen et al. (2022), based on the supposition that BMI might influence drug exposure, dosing and toxicity, there was no significant difference in the BMI categories, although it was not associated with PFS (*p* = 0.59) [30]. Wu et al. (2023) conducted a large real-world multicenter study including 397 Chinese patients with HR+/HER2− advanced breast cancer treated with palbociclib plus ET. The study found no significant association between BMI and PFS (*p* = 0.9554), and BMI was also unrelated to treatment toxicity or dose modifications [33]. Fasching et al. (2023), in their prospective study in a cohort of 487 patients, found that BMI had no impact on PFS (HR = 0.987, 95% CI: 0.964–1.011, *p* = 0.2947) [34]. This was also further reported by the study of Zhang et al., 2024, in which no significant association was found between baseline BMI and worse OS (HR = 1.1, *p* = 0.7 for overweight, HR = 0.96, *p* = 0.9 for obese) or PFS (HR = 0.9, *p* = 0.8 for overweight, HR = 0.76, *p* = 0.5 for obese) measured at 3 months of treatment [36]. Obese patients (BMI ≥ 30 kg/m^2^) at the three-month mark of CDK4/6i treatment demonstrated significantly improved OS. This suggests that BMI measured during the initial stages of therapy may be a meaningful predictor of long-term survival.

### 3.3. Studies Assessing Additional Body Composition Parameters and Their Association with Survival Outcomes

Other studies utilized alternative body composition metrics beyond BMI, such as skeletal muscle area (SMA) and visceral adipose tissue (VAT), to better characterize their relationship with treatment response and survival outcomes in HR+/HER2− MBC patients receiving CDK4/6i.

Yücel et al. (2024) found no significant association between BMI and PFS (HR = 1.22, 95% CI: 0.57–2.59, *p* = 0.599). In contrast, body composition analysis via computed tomography revealed significant clinical correlations. Patients with low SMA index or sarcopenia experienced significantly shorter PFS compared to non-sarcopenic patients (9.0 vs. 19.6 months; *p* = 0.005). Similarly, a low VAT index was associated with poorer outcomes (9.3 vs. 20.4 months; *p* = 0.033) [29]. In univariate analysis, both low SMA index (HR = 3.89, 95% CI: 1.35–11.25; *p* = 0.012) and low VAT index (HR = 2.15, 95% CI: 1.02–4.53; *p* = 0.042) were significantly associated with reduced PFS, while multivariate analysis identified sarcopenia as the only independent predictor of poor PFS (HR = 3.99, 95% CI: 1.38–11.54; *p* = 0.011) [29]. These results highlight that BMI alone is not a reliable prognostic marker, whereas sarcopenia and low visceral adiposity serve as potential early predictors of poorer outcomes in patients treated with CDK4/6i.

Franzoi et al. (2020) conducted a retrospective analysis performing computed tomography-based evaluation of body composition at baseline and during treatment to assess the prognostic relevance of skeletal muscle and adipose tissue indices. Sarcopenia at baseline significantly reduced PFS relative to non-sarcopenic status (9.6 months vs. 20.8 months; *p* = 0.037). In contrast, a high visceral fat index appeared to have a protective effect, as these patients experienced a median PFS of 20.8 months compared to 10.4 months in the low visceral fat group (HR = 0.44). No significant differences in PFS were observed according to BMI categories (20.8 vs. 12.1 months; HR = 1.23, 95% CI: 0.50–2.87; *p* = 0.624), indicating that BMI alone was not predictive of survival [32]. Furthermore, no significant changes in muscle or fat composition were detected during therapy. The authors concluded that sarcopenia represents a potential early marker of poor prognosis, while visceral adiposity may be associated with improved outcomes, emphasizing that computed tomography-based analysis provides more refined prognostic information than BMI in patients treated with CDK4/6i.

## 4. Discussion

CDK4/6i have achieved unprecedented success in oncology over the last few years, demonstrating significant improvements in survival. Despite the proven efficacy of CDK4/6i, emerging evidence suggests that BMI may represent a predictive factor worth considering, potentially influencing treatment outcomes and prognosis in patients with HR+/HER2− MBC. In this systematic review, we evaluated the association between BMI and other body composition parameters and survival outcomes of patients treated with CDK4/6i.

Overall, the evidence from the included studies is mixed. Although some studies observed a positive association between higher BMI and improved survival outcomes, most reported no meaningful correlation. Notably, some findings highlighted body composition parameters such as sarcopenia and visceral adiposity as superior indicators of treatment response and prognosis compared to BMI alone.

In line with previously published research, the findings of this review confirm that BMI does not consistently predict treatment outcomes in BC. In the metastatic setting, the prognostic value of BMI remains contentious. Emerging data in MBC suggest an “obesity paradox” where an elevated BMI may confer a survival advantage. A finding that contradicts established data in early-stage BC [5]. In early stage HR+/HER2− disease, obesity is a well-documented risk factor associated with significant reductions in DFS (HR = 1.26, 95% CI: 1.13–1.41) and OS (HR = 1.39, 95% CI: 1.20–1.62) [37]. Nevertheless, in our systemic literature review, the apparent “obesity paradox” was not observed. Two studies conducted in early-stage BC, one in the neoadjuvant and the other in the adjuvant setting5showed no significant relationship between BMI and treatment outcomes [9,25].

As molecularly targeted agents, CDK4/6i are administered to patients with specific tumor biological profiles, primarily HR+/HER2− disease and preserved Rb function. In addition to their central role in cell cycle arrest, cyclins 4 and 6 are also implicated in several metabolic processes, including adipogenesis, gluconeogenesis, and mitochondrial and muscle metabolism [18]. Therefore, BMI could potentially affect the therapeutic response to CDK4/6i. Adipose tissue exerts critical endocrine functions through estrogen production and secretion of adipokines such as leptin and adiponectin. In low-BMI states, depletion of adipose-derived hormones and altered insulin/*IGF-1* signaling may impair pathways that interface with the cyclin D–CDK4/6–Rb axis, thereby reducing CDK4/6i effectiveness [38].

Several interrelated biological pathways may explain the correlation between low BMI and sarcopenia and reduced treatment efficacy. Low BMI and decreased lean body mass can significantly alter the pharmacokinetics of lipophilic CDK4/6i, modifying hepatic metabolism and apparent volume of distribution. Roncato et al. demonstrated that patients with lower BMI (<25) often exhibit higher plasma drug concentrations, which may increase the risk of hematologic toxicities such as neutropenia, potentially resulting in treatment discontinuation or poor adherence [24]. Similarly, the PALLAS trial analysis suggested that body mass influenced tolerability and adherence to palbociclib therapy through side effects, dose reductions and relative dose intensity [25]. Lower BMI is associated with more frequent treatment modifications and poorer adherence, whereas higher BMI patients, experiencing fewer toxicities, often maintain consistent therapy, factors that may contribute to improved PFS [24].

In addition to BMI, two studies included in this review assessed VAT and SMA as specific body composition parameters related to treatment outcomes [29,32]. Evidence suggests that the VAT index is a more reliable predictor of PFS than BMI. For example, Yucel et al. [29] observed improved PFS in patients with higher visceral adiposity (reversed HR = 0.465, *p* = 0.042), while Franzoi et al. [32] demonstrated that elevated SMA and VAT indices nearly doubled median PFS. These findings contrast with the lack of significant results for BMI in the same cohorts. The term sarcopenia refers to a loss of skeletal muscle mass and strength [39]. Furthermore, the prevalence of sarcopenia in underweight patients likely contributes to their poorer outcomes [29,39,40]. Collectively, these data suggest that detailed body composition metrics provide clinical insights that BMI fails to capture. In addition, Kripa et al. reported that both higher VAT and higher SMA index were significantly associated with improved PFS (HR = 0.476, *p* = 0.008; HR = 0.687, *p* < 0.001), highlighting sarcopenia as a negative prognostic factor in HR+/HER2− MBC treated with CDK4/6i [40].

Although data on VAT and SMA were derived from relatively small cohorts, these parameters were assessed as complementary to BMI, particularly in light of the inconsistent prognostic impact of BMI alone. This has prompted increasing interest in alternative body composition metrics that may better reflect metabolic status in patients treated with CDK4/6i.

A major strength of this systematic review is the comprehensive inclusion of real-world studies and clinical cohorts evaluating the prognostic influence of BMI and body composition parameters in HR+/HER2− BC treated with CDK4/6i. The review integrates publications across multiple indexed sources that represent diverse geographic settings. While heterogeneity enhances external validity and relevance to routine oncology practice, it also limits direct comparability due to differences in sample size, assessment methods, and study design. Differences in patient populations were evident, with studies conducted in Asian cohorts often reporting distinct BMI distributions and metabolic profiles compared with Western populations, potentially influencing the observed associations between BMI and clinical outcomes. Several studies reported a higher incidence of hematologic toxicity in patients with lower BMI, including Asian populations [9,24]. Although dose reductions generally do not compromise efficacy, more frequent treatment modifications or reduced adherence in lower-BMI patients may negatively affect outcomes [24]. Conversely, patients with higher BMI tend to experience fewer adverse events and maintain treatment continuity, potentially contributing to improved PFS. Due to the timing of FDA approvals, many real-world studies predominantly included palbociclib, with less representation of ribociclib or abemaciclib among CDK4/6i regimens.

Additionally, the inclusion of studies employing image-based composition analysis, such as VAT and SMA, provide biologically meaningful insights that go beyond BMI. While few studies included more body composition indexes, routinely measuring them could result in a better assessment of toxicity risk, more so than a baseline DEXA Scan alongside the routine staging imagery, and could provide further information on body composition. The modest association observed between BMI and clinical outcomes, compared with the VAT index, may be partly explained by the BMI cut-offs used across studies. Most included analyses applied a BMI threshold around 25, potentially overlooking the impact of extreme BMI categories. Notably, Lammers et al. reported poorer OS among underweight patients (BMI < 18.5; HR = 1.45, *p* = 0.07) [26]. Further studies specifically addressing obesity (BMI ≥ 30) and underweight populations are needed to better clarify these relationships.

Despite the strengths, the review is limited by several methodological constraints inherent to the available science. The majority of included studies are retrospective observational cohorts, which are susceptible to selection bias and missing data that may influence the association between BMI and clinical outcomes. Only a limited number of prospective trials directly analyzed BMI as a predefined variable, making causal inference challenging, such as the one published by Pfeiler et al. [25]. As mentioned before, relying on BMI as a surrogate measure of body composition is imprecise in defining obesity as it cannot distinguish between lean and fat mass. Furthermore, follow-up duration and lines of treatment vary widely across the metastatic cohorts, and potential underreporting of adverse effects should be taken into account when interpreting the results.

We believe this review provides a robust and clinically relevant synthesis of current evidence linking BMI and body composition indexes to CDK4/6i treatment survival outcomes. Nonetheless, the predominance of retrospective designs and variability in anthropometric measurement highlights the need for standardized, prospective body composition-guided research to define optimized patient selection, supportive care strategies and dose adjustment protocols.

## 5. Conclusions

This systematic review offers a comprehensive synthesis of current evidence regarding the prognostic impact of BMI in patients with HR+/HER2− MBC. Our findings suggest that BMI as a standalone metric is an insufficient predictor of clinical outcomes or treatment response for those receiving CDK4/6i. These results underscore the critical need for more precise body composition assessments to better inform prognostic stratification and treatment strategies. The assessment of parameters such as anthropometric measurements and laboratory markers may help clarify the role of metabolism and nutritional status on breast cancer outcomes, with potential therapeutic implications. Furthermore, additional research is needed to elucidate the distinct impact of adiposity on outcomes in early-stage versus MBC. Future prospective trials should incorporate standardized body composition assessments, rather than relying solely on BMI, to better define how obesity and sarcopenia influence treatment efficacy. These insights will be vital for personalizing therapy in the HR+/HER2− patient population.

## Figures and Tables

**Figure 1 cancers-18-00081-f001:**
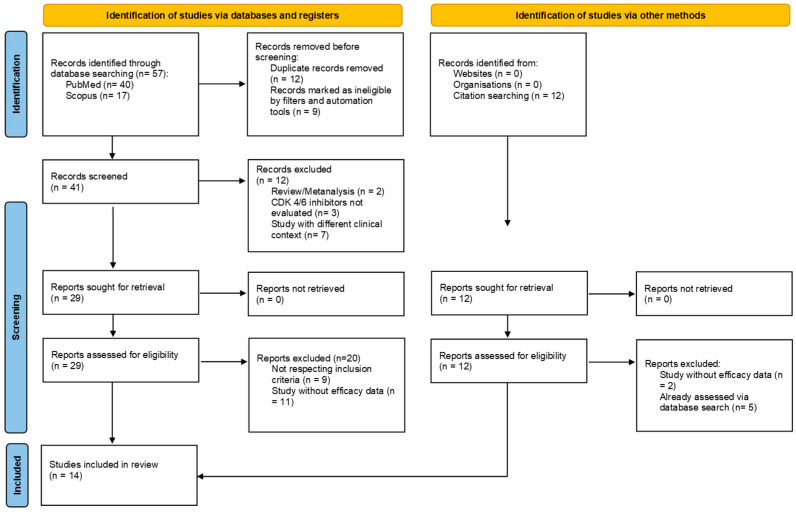
PRISMA flowchart of literature research and study selection.

**Table 1 cancers-18-00081-t001:** Quality and risk of bias assessment of the analyzed studies.

Study Title	S1	S2	S3	S4	C1	C2	O1	O2	O3	Total
Roncato, 2023 [24]	★	★	★	★	★	★	★	★		8
Pfeiler, 2023 [25]	★	★	★	★	★	★	★	★	★	9
Franzoi, 2021 [9]	★	★	★	★	★	★	★	★		8
Lammers [26]	★	★	★	★	★	★	★	★	★	9
Franzoi, 2022 [27]	★	★	★	★	★		★	★		7
Çağlayan, 2024 [28]	★	★	★	★	★		★	★		7
Yücel, 2024 [29]	★	★	★	★	★		★	★		7
Knudsen, 2022 [30]	★	★	★	★	★		★	★		7
Chen, 2024 [31]	★	★	★	★	★	★	★	★		8
Franzoi, 2020 [32]	★	★	★	★	★		★	★		7
Wu, 2023 [33]	★	★	★	★	★		★	★		7
Fasching, 2023 [34]	★	★	★	★	★		★	★	★	8
Shen, 2022 [35]	★	★	★	★	★	★	★	★		8
Zhang, 2024 [36]	★	★	★	★		★		★	★	7

★ indicates one point awarded for each satisfied quality criterion according to the NOS.

**Table 2 cancers-18-00081-t002:** Main characteristics of the included studies.

Author, Year	Type of Study	Country(ies)	Inclusion Period	Patient Number (n)	CDK 4/6 Inhibitor	Metrics	Criteria	Median BMI	Survival Outcmes	HR, 95% CI	*p* Value	Statistical Method(s)	Comment
Roncato, 2023 [24]	Prospective	Caucasian	2020–2022	134	Palbociclib	BMI	BMI ≥ 25 BMI < 25	NA	PFS	2.32 (1.09–4.95)	0.0332	Cox regression	BMI < 25 associated with shorter PFS
Pfeiler, 2023 [25]	Prospective	Multinational	2015–2018	5630	Palbociclib	BMI	25 > BMI ≥ 18.5 30 > BMI ≥ 25 BMI ≥ 30	26.7	iDFS	0.84 (0.63 to 1.12) 1.10 (0.82 to 1.49) 0.95 (0.69 to 1.30)	NA	Cox regression	No association of BMI with iDFS
Franzoi, 2021 [9]	Pooled analysis	Multinational	2014–2015	757	Abemaciclib	BMI	BMI ≥ 25 BMI < 25	25.5 (22.2–29.7)	PFS; ORR; CBR	1.03 (0.83–1.27); 0.73 (0.54–0.99); 0.73 (0.41–1.31)	0.81 0.04 0.28	Cox proportional hazard regression; log-rank tests	No association of BMI with PFS and CBR. Positive association of normal and underweight with ORR. No association between PFS and weight loss
Lammers, 2023 [26]	Retrospective	Netherlands	2007–2020	256	Not Specified	BMI	BMI ≤ 18.5 30 > BMI ≥ 25 BMI ≥ 30	NA	OS, PFS	OS: 1.45 (0.97–2.15); 0.99 (0.85–1.16); 1.04 (0.88–1.24); PFS: 1.05 (0.73–1.51); 0.90 (0.79–1.03); 0.88 (0.76–1.02)	OS:0.07; 0.93 0.62 PFS:0.81 0.14; 0.10	Cox proportional hazard regression; log-rank tests	No association of BMI with OS and PFS
Franzoi, 2022 [27]	Post-hoc analysis of RCT	Multinational	NA	222	Abemaciclib	BMI	BMI ≥ 25 BMI < 25	NA	Clinical and radilogcal response		Clinical reponse: 0.261; Radiological response: 0.366		No association of BMI with radiological and clinical response rate. No impact on Ki-67 % changes
Çağlayan, 2024 [28]	Retrospective	Turkey	2019–2021	116	Palbociclib, Ribociclib	BMI	25 > BMI ≥ 18.5 30 > BMI ≥ 25 BMI ≥ 30	NA	PFS	NA	0.02	Cox regresion model, log-rank test	Positive association of BMI with PFS
Yücel, 2024 [29]	Retrospective	Turkey	2018–2021	52	Palbociclib, Ribociclib	BMI VAT SMA	BMI ≥ 30 BMI < 30 VAT High VAT Low SMA High SMA Low	27.6	PFS	BMI 1.22 (0.57–2.59); VAT 2.15 (1.027–4.535) SMA 3.89 (1.353–11.25)	0.599 0.042 0.012	Cox proportional hazard regression	No association of BMI with PFS. High VAT index had longer mPFS. Patients with sarcopenia (low SMA index) had poorer mPFS
Knudsen, 2022 [30]	Retrospective	USA	2015–2021	222	Palbociclib Ribociclib Abemaciclib	BMI	BMI < 18.5 24.9 > BMI > 18.5 29.9 > BMI > 25 BMI ≥ 30	27.85	PFS	NA	0.59	Cox regression model, log-rank test	No association of BMI with PFS
Chen, 2024 [31]	Retrospective	Taiwan	2018–2023	340	Palbociclib, Ribociclib	BMI	BMI ≥ 25 BMI < 25	23.51	PFS	0.943 (0.907–0.980)	0.003	Cox proportional hazard model, log-rank test	Positive association of BMI with PFS
Franzoi, 2020 [32]	Retrospective	Belgium	2016–2019	50	Not specified	BMI VAT	BMI ≥ 25 BMI < 25 VAT High VAT Low		PFS	BMI 1.23 (0.5–2.87); VAT 0.44 (0.18–1.06)	BMI 0.592 VAT 0.041	Cox regression model	No association of BMI with PFS. Positive association of high visceral fat index with PFS
Wu, 2023 [33]	Retrospective	China	2016–2022	397	Palbociclib	BMI	BMI ≥ 24 BMI < 24	NA	PFS	NA	0.9554	Cox regression model	No association of BMI with PFS
Fasching, 2023 [34]	Prospective	Germany	2016–2020	481	Ribociclib	BMI	Not specified	NA	PFS	0.987 (0.964–1.011)	0.2947	Cox regression model	No association of BMI with PFS
Shen, 2022 [35]	Ambispective	China	2018–2020	190	Palbociclib	BMI	BMI ≥ 24 BMI < 24	22.96	PFS	0.98 (0.66–1.47)	0.928	Cox proportional hazard models	Positive association of BMI with PFS
Zhang, 2024 [36]	Retrospective	USA	2015–2023	221	Palbociclib, Ribociclib, Abemaciclib	BMI	BMI ≥ 30 30 > BMI ≥ 25 25 > BMI ≥ 18.5		OS, PFS	OS: 0.96; 1.10 PFS: 0.76 (0.35–1.66); 0.9 (0.42–1.94)	OS: 0.9 0.7 PFS: 0.5 0.8	Cox proportional hazard analysis	No association of BMI with OS and PFS. Positive association between OS and obese BMI after 3 months of treatment.

## Data Availability

The data generated or analyzed during this study are included in this published article or are available from the corresponding author on reasonable request.

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
