# Peer review of "The Prognostic and Predictive Value of Body Mass Index in Patients with HR+/HER2− Breast Cancer Treated with CDK4/6 Inhibitors: A Systematic Literature Review"

_cancers, 2025, doi:10.3390/cancers18010081_

Round 1
Reviewer 1 Report
Comments and Suggestions for Authors
Dear Authors,
First of all, congratulations for your interesting work. I hope that my hints will help you in the next steps of improvement and the final manuscript will be really valuable for the readers. Your systematic review synthesizes evidence on whether BMI predicts clinical outcomes in HR+/HER2- metastatic breast cancer patients treated with CDK4/6 inhibitors. It demonstrates inconsistent associations and highlights that BMI is an inadequate marker, while CT-based body composition measures—particularly sarcopenia and visceral adiposity—provide more meaningful prognostic information.
There are several typoes and many punctuation mistakes (such as double space, double dot or no at all) and some typos - even if they do not change the value of the manuscript, I'd like to urge you to correct these imperfections. Some sentences are overly long and could be simplified for readability. Also, there are a lot of grammatical errors - please, revise the manuscript.
Moreover, gene names should be written in italics, in contrast to the protein names, according to the rules of genetic consensus. Please, familiarise yourself with the rules and change the manuscript accordingly. Examples of rules summary can be found on websites such as: https://www.gmb.org.br/geneprotein-nomenclature-guidelinesor https://academic.oup.com/molehr/pages/Gene_And_Protein_Nomenclature
Revise also the inconsistent spelling of technical terms: CDK 4/6, CDK4/6, CDK 4/6i, CDK4/6i.
Excessive heterogeneity without structured analysis: the review highlights heterogeneity but does not adequately analyze sources of heterogeneity (ethnicity, treatment line, CDK4/6i type, BMI cutoffs, etc.). Can you add subgroup analyses or a narrative comparison (e.g., Asian vs Western cohorts, palbociclib-only vs mixed CDK4/6i). Consider also adding a summary table comparing methodological differences.
Consider creating some schemes or graphical presentations of the discussing elements.
Finally, CDK4/6i drugs are usually given to patients posessing specific genetic variants. They are typically a moleculary-targeted therapies. Yet, there is nothing about this important point in your manuscript. Please amend your document, at least in the discussion and introductory parts.
Reviewer 2 Report
Comments and Suggestions for Authors
The work included studies that are very heterogeneous (14 studies, of which only 2 are
prospective)
Four excluded studies are mentioned as having been removed using an automated tool,
without specifying which tool was used
The paper claims to focus on the metastatic population, but among the 14 included
studies, 2 are on early-stage disease
The evaluation of parameters other than BMI (VAT and SMA) is available only for a rather
limited sample size (50 patients in one study and 50 patients in another), making it difficult
to draw conclusions
Overall, the study has some potential interest, but in fact, by the end of the reading it does
not seem to add anything new
The presence of grammatical errors and typos mandates language revision.
Need for improvement
Round 2
Reviewer 1 Report
Comments and Suggestions for Authors
Dear Authors, thank you very much for your corrected document. Good luck in your future endavours!
Reviewer 2 Report
Comments and Suggestions for Authors
AUthors addressed issues that had been raised. Overall, the study does not clarify the role, if any, of BMI